# Research on Multi-Stage Post-Occupancy Evaluation Framework of Community Comprehensive Elderly Care Service Facilities under the Public-Private Partnership Mode—A Case Study of China

**Lijun Lin [1], Lin Zhang [2,\*], Shuai Geng [2], Yulin Zhao [2] and Yuanyuan Tian [2]**

1   School of Science, Shandong Jianzhu University, Jinan 250101, China; linlijunxueshu@163.com
2   School of Management Engineering, Shandong Jianzhu University, Jinan 250101, China; gengshuai18@sdjzu.edu.cn (S.G.); 15397721566@163.com (Y.Z.); tianyuanyuan19@sdjzu.edu.cn (Y.T.)
\*   Correspondence: zhanglinxueshu@163.com

**Abstract:** The key to whether elderly individuals in the community can enjoy their later years peacefully lies in the service capabilities of community comprehensive elderly care service facilities (CCECSF) under the Public-Private Partnership (PPP) mode. To maintain a high level of service capability in community comprehensive elderly care service facilities under PPP mode, scientific evaluation of the effectiveness of these facilities is equally crucial. This article first constructs a post-occupancy evaluation index system of community comprehensive elderly care service facilities under PPP mode that includes three evaluation attributes and fifteen evaluation criteria based on the Chinese culture and lifestyle habits. Regarding the issue of direct users being unable to directly participate in evaluations, the uncertainty in evaluation information, and the volatility of evaluation results, a multi-stage post-occupancy evaluation model is constructed based on probabilistic linguistic term set, TOPSIS model and multi-stage decision theory. The above post-occupancy evaluation index system and evaluation model together constitute a multi-stage post-occupancy evaluation framework for community comprehensive elderly care service facilities under PPP mode. The outcomes of the case study indicate that the post-occupancy evaluation index system can offer a scientifically guided approach for evaluating the service level of community comprehensive elderly care service facilities under the PPP mode; meanwhile, the multi-stage evaluation model can enable direct user participation in the post-evaluation of facility usage and improve the robustness and reduce the fluctuation of the evaluation results, so as to improve the scientificity of the evaluation results.

**Keywords:** community comprehensive elderly care service facility; Public-Private Partnership model; multi-stage post-occupancy evaluation; probabilistic linguistic term set

## 1. Introduction

The escalating global aging crisis has prompted a widespread discourse on eldercare provision. Within the framework of worldwide demographic aging, elderly individuals increasingly favor community or familial care due to constraints related to pension expenses, personal preferences and quality considerations [1]. Nations experiencing demographic shifts toward aging populations are exploring strategies to facilitate community-based eldercare, aiming to alleviate governmental burdens. Consequently, community-centric elderly care is emerging as the prevailing paradigm in response to the imperative of optimizing resources and enhancing care efficacy [2,3]. To optimize community elderly care, Scholars Puustinen and Jonna believe that when providing comprehensive elderly care services in communities, the views of the elderly on how they wish to plan and provide care and services should be fully considered, so that they can live an independent and fulfilling life [4]. The post-occupancy evaluation of the community comprehensive elderly

care service facility (CCECSF) allows for better tailoring of services to meet the expectations of the elderly, ensuring the provision of higher-quality care. Consequently, a scientifically grounded assessment of these facilities is crucial.

In addition, due to the insufficient status of China's pension system, relying solely on the government is inadequate to sustain the operation of community comprehensive elderly care service facilities [5]. Therefore, a common practice is to allow social capital to construct these facilities through the Public-Private Partnership (PPP) model, granting them franchise rights for operation. However, current post-evaluation studies on community comprehensive elderly care service facilities do not incorporate the PPP mode, despite its crucial impact on the post-evaluation of such facilities. The choice of investment model significantly influences the effectiveness of facility utilization. This is attributed to the profit-driven nature of capital, where increasing fees and reducing operational costs are the most direct ways to enhance profitability. Elevated fees may affect cost-effectiveness, while decreased operational costs imply lower service quality. Consequently, this paper focuses on investigating community comprehensive elderly care service facilities under the PPP mode. It aims to establish a post-evaluation framework to ensure that private capital can consistently provide high-quality services for the elderly in the community. Post-evaluation research constitutes a subset of evaluation studies, which can be categorized into two types: empirical research and evaluation framework research.

In terms of empirical research, the research method is to build a post-evaluation index system based on the characteristics of elderly care service facilities in a certain region and conduct empirical research on facilities in the region through data envelopment analysis [6] to find key factors or constraints and propose improvement strategies. For example, Baldwin and Richard studied some elderly care institutions in Australia by establishing an evaluation system and found that factors such as the scale and facility completion rate of elderly care institutions would have a great impact on the service quality of elderly care institutions [7]. However, their research findings have certain limitations and may not be fully applicable to China. Due to the cultural differences between the East and the West, China and Australia have different traditions in elderly care. Foreign elderly people choose to live alone or in nursing homes, while Chinese elderly people prefer to live at home with their children. Therefore, the above research results cannot be fully applicable to China.

Different from empirical research, the focus of evaluation framework research is not the analysis of implementation, but whether the evaluation index system and evaluation model are scientific enough, that is, whether the relevant criterion system contains all the factors that need to be considered, whether the evaluation model can scientifically determine the weight of criteria and evaluators, whether it can scientifically aggregate data from different aspects and whether the uncertainty in the evaluation process can be effectively dealt with to make the evaluation results have a certain robustness. The study of evaluation framework is the basis of empirical research to ensure the scientific results of empirical research [8].

However, the existing research predominantly consists of empirical studies. Simultaneously, in studies concerning the evaluation framework of community comprehensive elderly care service facilities, the primary operational approach of community comprehensive elderly care service facilities has been a government monopoly. However, due to economic downturns and fiscal constraints, the PPP model is poised to become the primary operational mode for community comprehensive elderly care service facilities. The variance in operational models, whether PPP or government monopoly, significantly impacts the service quality of community comprehensive elderly care service facilities [9]. Therefore, the current research gap lies in how to conduct a scientific post-occupancy evaluation of the efficacy of community comprehensive elderly care service facilities under the PPP mode.

The purpose of this study is to construct a post-occupancy evaluation framework for community comprehensive elderly care service facilities under PPP mode. In the following chapters, the advantages and disadvantages of the existing evaluation index

system and evaluation model of community comprehensive elderly care service facilities will be introduced and the innovative points of this article will be proposed.

## 2. Literature Reviews

### 2.1. Literature Review of Post-Occupancy Evaluation Index System of Community Comprehensive Elderly Care Service Facilities under PPP Mode

Service quality is an issue that must be considered in the post evaluation of community comprehensive elderly care service facilities. In China, Yang Qianwen contends that the optimal outcome in home-based elderly care services lies in service effectiveness, specifically, the satisfaction of consumers with service quality [10]; Bai Jie categorizes community living facilities into three groups—health care, daily services and entertainment—aligned with residents' living needs. She further advocates for specific service functionalities within each category under the home-based elderly care model [11]; addressing specific physiological and psychological requirements of the elderly, Sun Fallow Zhi structured an evaluation framework based on user needs. This framework comprises seven dimensions: safety, comfort, privacy, accessibility, physical and cognitive support, social support and a sense of belonging, offering a post-occupancy assessment of community comprehensive elderly care service facilities [12]; Compared with China, Europe, the United States, Japan and South Korea entered the aging society earlier, and the exploration of the pension model was earlier and more mature. In the post-occupancy evaluation of elderly care facilities, the focus is on quality of life, happiness and satisfaction, or services for special groups, such as the elderly with cognitive disorders [13]. In summary, the service quality of community comprehensive elderly care service facilities mainly reflects whether it can meet the needs of the elderly, such as user's general demand, user's physical demand and user's cognitive demand [12,14].

In addition, the cost of using community comprehensive elderly care service facilities also has a significant impact on service quality. Liao Chu Hui et al. revealed that operational costs, the health of elderly care service functions, and the actual needs of spiritual and cultural aspects significantly influence service quality in such areas [15]. Rita Yi Man Li et al. found that the elderly typically downsize from larger homes and relieve their financial needs [16]. The cost-effectiveness of services constitutes a crucial consideration. The user cost includes the following three parts: health care cost, daily service cost and recreational activity cost [11,15].

The aforementioned research provides a good foundation for the study of the text, but the existing problem is that the impact of PPP mode on the post-occupancy evaluation of community comprehensive elderly care service facilities in communities is not considered. Under the PPP mode, it is the government that grants special management rights to social capital. Hence, the inefficacy of community comprehensive elderly care service facilities usage can significantly impede the governmental political performance. Consequently, any post-occupancy evaluation of community comprehensive elderly care service facilities within the PPP mode should incorporate the assessment of facilities' impact on governmental political efficacy [17]. The government performance of community comprehensive elderly care service facilities under PPP mode includes the following two parts: community comprehensive elderly care coverage and service effect [10,17,18].

In summary, a post-occupancy evaluation index system for community comprehensive elderly care service facilities under PPP mode can be constructed based on the three attributes of user demand, user cost and government performance, as well as their evaluation criteria. The specific content can be found in Section 3.

### 2.2. Literature Review of Post-Occupancy Evaluation Models of Community Comprehensive Elderly Care Service Facilities

Previous studies on post-occupancy evaluation of community comprehensive elderly care service facilities mainly focused on empirical research, and the data were mainly statistical data or survey data. The method of data aggregation was the weighted average method, and the research focus was on the evaluation index system rather than the

evaluation model [19]. But in the evaluation framework research, the evaluation model is also the focus of research. In order to obtain more realistic and scientific evaluation results, the post-occupancy evaluation of community comprehensive elderly care service facilities under PPP mode needs to involve the elderly using the facilities directly in the evaluation, and it also needs to evaluate the sustainability of the use effect of the facilities. From the perspective of decision science, post-occupancy evaluation of community comprehensive elderly care service facilities under PPP mode involves large-group, multi-stage decision-making. It necessitates aggregating evaluation data from over 20 evaluators into a comprehensive metric and consolidating evaluation values from different time intervals to ensure the sustainability of evaluation results. The first challenge is the uncertainty in evaluation information, where assessors' hesitations lead to value fluctuations, compromising robustness. Another challenge involves aggregating preferences from a large user base, often exceeding 100 individuals. Additionally, there's the problem of the use effect of sustainable evaluation. Post-occupancy evaluation results of community comprehensive elderly care service facilities under PPP mode should be sustainable results, and how to scientifically aggregate the post-occupancy values at different evaluation stages is a problem that needs to be studied.

For the uncertainty in evaluation information, fuzzy mathematics is frequently employed to address the uncertainty inherent in evaluation data. Such as fuzzy mathematics [20], intuitional fuzzy numbers [21], interval intuitional fuzzy numbers [22], hesitant fuzzy numbers [23], Pythagorean intuitional fuzzy numbers [24], probabilistic linguistic term set (PLTS) [25], etc. However, the expression of numbers is not in line with the user's expression habits for evaluation, especially for the elderly, so the expression of numbers will cause more uncertainty. For example, some elderly people use 0.98 to represent "good", while some old people use 0.8 to represent "good"; the more complex the expression form of fuzzy mathematics, the greater the uncertainty. Therefore, for the elderly, it is better to use linguistic terms that they are more familiar with to express their opinions, but the uncertainty processing ability of linguistic terms is not as good as that of fuzzy mathematics. Fortunately, PLTS can solve this problem. Contrasted with fuzzy mathematics, linguistic terms in PLTSs, such as "good" and "bad", offer greater convenience for elderly individuals in articulating their opinions. Probability associated with linguistic terms addresses the inherent uncertainty when elders express their preferences.

Because of the aggregation of evaluation preferences of large groups, the advantage of using PLTS as the mathematical basis of the evaluation model is that it is easier to aggregate evaluation data of large groups. In the process of constructing the evaluation value of PLTS, it can be considered that the importance of each evaluator is the same. Therefore, the probability of a linguistic term in a PLTS is the number of people who choose the linguistic term divided by the total number of people. Therefore, it is not necessary to cluster and group first, and then calculate the comprehensive evaluation value of the subgroup, as the general large group decision model, and then aggregate the comprehensive evaluation value of the subgroup [26].

While PLTS proves effective in managing uncertainty during the post-implementation evaluation of community comprehensive elderly care service facilities and facilitates efficient aggregation of large group evaluations, its computing complexity poses challenges. The intricate nature of PLTS expression hinders clear classification of community comprehensive elderly care service facilities service levels based on comprehensive evaluation values. Such as the comprehensive evaluation value is PLTS $\{S_{-2}(0.4), S_{-1}(0.2), S_0(0.1), S_0(0.1), S_1(0.1), S_2(0.2)\}$ according to the linguistic terms and symbols in Table 1 then the grade of community comprehensive elderly care service facilities needs to be obtained through complex calculation. Hence, PLTS-derived evaluation values necessitate aggregation into a real-number-based comprehensive evaluation using a scientific aggregation model.

**Table 1.** Value range, linguistic terms and symbols.

| Valuation Range | Description | Symbol |
|---|---|---|
| [0, 0.11] | Horrible | $s_{-4}$ |
| (0.11, 0.22] | Very poor | $s_{-3}$ |
| (0.22, 0.33] | Poor | $s_{-2}$ |
| (0.33, 0.44] | Relatively poor | $s_{-1}$ |
| (0.44, 0.55] | General | $s_0$ |
| (0.55, 0.66] | Relatively good | $s_1$ |
| (0.66, 0.77] | Good | $s_2$ |
| (0.77, 0.88] | Very good | $s_3$ |
| (0.88, 1] | Excellent | $s_4$ |

Due to its ability to effectively handle uncertainty in evaluation information while meeting the expression habits of elderly evaluators, PLTS is more suitable as the mathematical foundation for post-occupancy evaluation models because it is conducive to the aggregation of evaluation information for large groups.

In terms of evaluation models, there are two types: classification models and multi-attribute decision models.

Classification models include: Linear Regression Model [27], Support Vector Machine [28], Neural Network Class Model [29] and Machine Learning [30]. Classification models are mainly applied to linearly separable data, that is, the data can be divided into two categories through linear classifiers (such as straight lines and planes, etc.) [29]. The classification model needs to build a corresponding classification model based on a large number of classification instance data. However, due to data of community comprehensive elderly care service facilities not being shared, it is difficult to find a large amount of data, and the above model cannot handle the complex mathematical expression data, so it cannot be used to evaluate the usage effect of community comprehensive elderly care service facilities under PPP mode.

The common multi-attribute decision-making models include the Elimination Et Choice Translating Reality (ELECTRE) method [31], Preference Ranking Organization Method For Enrichment Evaluations (PROMETHEE) method [32], Analytic Hierarchy Process (AHP) method [33], Technique for Order Preference by Similarity to Ideal Solution (TOPSIS) method [34] and so on. However, the weighted sum method yields a PLTS outcome, and both Elimination Et Choice Translating Reality (ELECTRE) and Preference Ranking Organization Method For Enrichment Evaluations (PROMETHEE) methods are unsuitable for evaluation. Consequently, the Technique for Order Preference by Similarity to Ideal Solution (TOPSIS) method emerges as the optimal choice for post-occupancy evaluation of community comprehensive elderly care service facilities. The Technique for Order Preference by Similarity to Ideal Solution (TOPSIS) method transforms PLTS evaluation values into real numbers by calculating the distances between the evaluated community comprehensive elderly care service facilities project and the positive and negative ideal solutions, facilitating subsequent community comprehensive elderly care service facilities categorization. Nonetheless, issues arise in the application of Technique for Order Preference by Similarity to Ideal Solution (TOPSIS) in post-occupancy community comprehensive elderly care service facilities evaluation. Firstly, if only one comprehensive elderly care service facility participates, it remains unassessed. Secondly, the variance in positive and negative ideal solutions across different time evaluation stages impedes comparisons of post-occupancy results.

For addressing multi-stage evaluation issues, the application of multi-stage decision theory is warranted. In this theory, stage decision weights are established to amalgamate decisions from different stages into a comprehensive result. The most prevalent method for setting stage weights in this theory is subjective judgment, yet it is deemed excessively subjective. Apart from subjective judgment, objective weighting methods based on arithmetic series, geometric series, and normal distribution have been proposed [35]. These

methods are objective weighting approaches, disregarding subjective factors by leveraging differences among data. However, the rationality of evaluation is a subjective concept, and neglecting subjective intent in stage weight assignment may render results less acceptable. Fortunately, researcher Xiuli Geng has introduced an information entropy method based on expert stage preferences [36]. This method determines stage weights based on information entropy when evaluators possess explicit stage preferences, mitigating the arbitrariness in weight assignment to some extent and offering a degree of objectivity. Thus, this study will employ this method for stage weight determination.

*2.3. Motivation and Innovation of This Paper*

This paper addresses the scientific evaluation of the utilization of community comprehensive elderly care service facilities under the PPP model. It aims to construct a comprehensive post-occupancy evaluation index system considering user demand, user cost and government performance. The objective is to provide a clear direction for the scientific evaluation of these facilities. To handle the uncertainty in post-occupancy evaluation information and aggregate large group evaluation values, the paper employs PLTS.

In the multi-stage evaluation model proposed in this paper, aggregation of current-stage assessment information employs the Technique for Order Preference by Similarity to Ideal Solution (TOPSIS) method under fixed positive and negative ideal solutions as the aggregation model. This approach transforms PLTS-based assessment information into real-numbered evaluation values, thereby reducing computational complexity in subsequent steps without compromising decision information. For setting stage weights at different time stages, the information entropy method based on expert stage preferences is employed to ensure objectivity while accommodating decision-makers' subjective preferences. Building upon the evaluation index system and the multi-stage evaluation model, a framework for post-evaluation of community comprehensive elderly care service facilities under the PPP model is established, facilitating a scientific evaluation. The specific innovations are detailed as follows:

- The post-occupancy evaluation index system of community comprehensive elderly care service facilities under PPP mode has been constructed to guide scientific evaluation;
- PLTS is employed to express the evaluation value, effectively managing uncertainty and facilitating the efficient aggregation of evaluations from large groups. This ensures scientifically sound evaluation results.
- Construct a multi-stage post-occupancy model for the community comprehensive elderly care service facilities to achieve a sustainable evaluation of community comprehensive elderly care service facilities.

## 3. Post-Occupancy Evaluation Index System of Community Comprehensive Elderly Care Service Facility under the PPP Mode

As per the literature review in Section 2, the assessment of community comprehensive elderly care service facilities' post-occupancy under the PPP mode involves examining three key aspects: user demand, user cost, and government performance. User needs mainly reflect whether the community comprehensive elderly care service facilities can meet the needs of users, such as universal needs, physical needs and cognitive needs; the user cost is mainly to reflect whether the community comprehensive elderly care service facilities are cost-effective. The government performance is mainly reflected in the PPP mode, whether the community comprehensive elderly care service facilities can meet the political performance of the government, such as coverage and comprehensive service effect. The specific criteria and their explanations are shown in Table 2.

**Table 2.** The post-occupancy evaluation index system of Chinese community comprehensive elderly care service facilities under the PPP mode.

| Attributes | Criterion | Sub-Criterion | Criterion Property | Explanations | Data Sources |
|---|---|---|---|---|---|
| user's demand (A1) | user's general demand (C11) | privacy (C111) | qualitative/positive | The degree to which the facility allows the user to maintain privacy, such as private space without unauthorized entry by outsiders [12,14]. | public |
| | | personalization (C112) | qualitative/positive | The construction and layout of facilities should fully refer to the views of users, such as the layout of the room [12,14]. | public |
| | | self-choice control (C113) | qualitative/positive | Users should have the right to choose and control the service facilities, such as free choice of dining, washing time, free control of toilet, bath equipment use or not. Rather than a fixed period of time, the use of such facilities is permitted only [12,14]. | public |
| | | social support (C114) | qualitative/positive | Refers to the extent to which service facilities can allow users to communicate in them, such as the space to meet the communication conditions of many people [12,14]. | public |
| | user's physical demand (C12) | safety (C121) | qualitative/positive | Refers to the extent to which the facility can maintain or promote a healthy and safe environment and prevent accidents, such as non-slip on the ground, emergency buttons and safety switches in the room [12,14]. | public |
| | | comfort (C122) | qualitative/positive | Refers to the degree of comfort in the building to the sound, air, temperature and other environment, such as the degree of cleanliness, no unpleasant smell [12,14]. | public |
| | | physical support (C123) | qualitative/positive | Physical support refers to the degree to which the building allows the elderly in need to maintain their independence, such as providing handrails along the route [12,14]. | public |
| | user's cognitive demand (C13) | cognitive criterion (C131) | qualitative/positive | Cognitive criterion means that the building allows users to choose the way of activity and life, without limiting the degree of user choice, such as the easily recognized entrance to the activity room [12,14]. | public |
| | | sense of belonging (C132) | qualitative/positive | Sense of belonging refers to the degree to which the facility space design provides a sense of community, such as including space for community public activities, and the user's familiarity with the facility environment [12,14]. | public |
| user cost (A2) | health care (C21) | health care cost (C211) | quantitative/negative | The use of diagnostic and treatment equipment, health care equipment and other costs accounted for the proportion of all costs [11]. | market |
| | daily service (C22) | daily service cost (C221) | quantitative/negative | The use of transportation equipment, elevator equipment maintenance and other costs account for the proportion of all costs [11]. | market |
| | recreational activity (C23) | recreational activity cost (C231) | quantitative/negative | The proportion of the cost of using recreational facilities and fitness equipment provided in the community in all expenses [11]. | market |
| government performance (A3) | Community comprehensive elderly care coverage (C31) | Coverage rate (C311) | quantitative/positive | The proportion of the elderly who adopt the community comprehensive elderly care service facility in the total number of the elderly [3]. | statistics |
| | service effect (C32) | The timeliness of service personnel (C321) | quantitative/positive | Whether the service personnel employed by the service enterprises generated by government bidding can provide services to consumers in a timely manner [37]. | expert |
| | | Professionalism of service personnel (C322) | quantitative/positive | Whether the service personnel employed by the service enterprises generated by government bidding can provide professional services to consumers [37]. | expert |

Note: The coverage rate of the elderly in the community comprehensive care service = the number of the elderly in the local community comprehensive care service model/the total number of local elderly. Sub-criterion can be expressed in a way that the public can understand, depending on the specific situation.

## 4. Post-Occupancy Evaluation Model of Community Comprehensive Elderly Care Service Facility under PPP Mode

*4.1. Operation Rules of PLTS*

Before building the model, we first introduce the basic operating rules of PLTSs, which are as follows:

**Definition 1 ([38]).** *Given a linguistic term set* $S = \{s_\alpha | \alpha = -\tau, \ldots, -1, 0, 1, \ldots, \tau\}$*, then the PLTS term set can be defined as follows:*

$$L(p) = \left\{ s^{(l)} \left( p^{(l)} \right) \middle| s^{(l)} \in S_2, p^{(l)} > 0, l = 1, 2, \ldots, \#L(p), \sum_{l=1}^{|L(p)|} p^{(l)} \leq 1 \right\}$$

*where* $s^{(l)} \left( p^{(l)} \right)$ *is the l-th element in a PLTS, which contains a linguistic term and the probability of the linguistic term.* $\#L(p)$ *represents the number of elements in a PLTS term set.*

Unlike linguistic variable operations, PLTS operations require sorting the elements in the set first, as detailed in Definition 3.

**Definition 2 ([38]).** *Assuming that a given a set of PLTS* $L(p) = \left\{ s^{(k)} \left( p^{(k)} \right) \middle| k = 1, 2, \ldots, \#L(p) \right\}$*,* $I \left( s^{(k)} \right)$ *is a subscript of* $s^{(k)}$*, If* $s^{(k)} \left( p^{(k)} \right) (k = 1, 2, \ldots, \#L(p))$ *is arranged in descending order based on* $I \left( s^{(k)} \right) p^{(k)} (k = 1, 2, \ldots, \#L(p))$*, then* $L(p)$ *is called an ordered PLTS.*

**Definition 3 ([38]).** *Let* $L_1(p)$ *and* $L_2(p)$ *a set of PLTSs for the two order* $L_1(p) = \left\{ s_1^{(k)} \left( p_1^{(k)} \right) \middle| k = 1, 2, \ldots, \#L_1(p) \right\}$ *and* $L_2(p) = \left\{ s_2^{(k)} \left( p_2^{(k)} \right) \middle| k = 1, 2, \ldots, \#L_2(p) \right\}$*, then*

$$L_1(p) \bigoplus L_2(p) = \cup_{s_1^{(k)} \in s_1(p), s_2^{(k)} \in s_2(p)} \left\{ p_1^{(k)} s_1^{(k)} \bigoplus p_2^{(k)} s_2^{(k)} \right\} \tag{1}$$

$$L_1(p) \bigotimes L_2(p) = \cup_{s_1^{(k)} \in s_1(p), s_2^{(k)} \in s_2(p)} \left\{ \left( s_1^{(k)} \right)^{p_1^{(k)}} \bigoplus \left( s_2^{(k)} \right)^{p_2^{(k)}} \right\} \tag{2}$$

$$\lambda L(p) = \cup_{s^{(k)} \in L(p)} \lambda p^{(k)} s^{(k)}, \lambda \geq 0 \tag{3}$$

$$(L(p))^\lambda = \cup_{s^{(k)} \in L(p)} \left\{ \left( L^{(k)} \right)^{\lambda p^{(k)}} \right\} \tag{4}$$

The $s_1^{(k)}$ and $s_2^{(k)}$ are the k-th linguistic terms in the $L_1(p)$ and $L_2(p)$, respectively, $p_1^{(k)}$ and $p_2^{(k)}$ are the probability of the k-th linguistic terms of $L_1(p)$ and $L_2(p)$, respectively, $\#L(p)$ is the number of linguistic terms in $L(p)$.

**Definition 4 ([38]).** *Suppose S is the linguistic term set,* $L(p) = \left\{ s^{(l)} (p^{(l)}) \middle| s^{(l)} \in S, l = 1, 2, \ldots, L \right\}$ *is a PLTS based on S, then the concentration degree of* $L(p)$ *can be calculated as follows:*

$$cd(L(p)) = 1 + \sum_{l=1}^{L} p^{(l)} \log_2 \left( 1 - \frac{\left| I\left(s^{(l)}\right) - I(E(L(p))) \right|}{I(d_{lts})} \right) \tag{5}$$

**Definition 5 ([38]).** *Suppose S is the linguistic term set,* $L(p) = \left\{ s^{(l)} (p^{(l)}) \middle| s^{(l)} \in S, l = 1, 2, \ldots, L \right\}$ *is a PLTS based on S, then the deviation degree of* $L(p)$ *can be calculated as follows:*

$$dd(L(p)) = -\sum_{l=1}^{L} p^{(l)} \log_2 \left( 1 - \frac{\left| I\left(s^{(l)}\right) - I(E(L(p))) \right|}{I(d_{lts})} \right) \tag{6}$$

where $I\left(s^{(l)}\right)$ is the subscript of $s^{(l)}$ linguistic term, $I\left(d_{lts}\right)$ is the difference between the largest linguistic term and the smallest linguistic term subscript in the linguistic term set, and $I\left(E\left(L\left(p\right)\right)\right)$ is the expected value of the linguistic term subscript in the PLTS $L\left(p\right)$.

**Definition 6** ([39]). *Given a linguistic term set $S = \{S_\alpha | \alpha = -\tau, \ldots, -1, 0, -1, \ldots, \tau\}$; $L_1\left(p\right) = \left\{s_\alpha^{(l)}\left(p_\alpha^{(l)}\right) \middle| s_\alpha^{(l)} \in S_1, l = 1, 2, \ldots, \#L_1(p)\right\}$ and $L_2\left(p\right) = \left\{s_\beta^{(l)}\left(p_\beta^{(l)}\right) \middle| s_\beta^{(l)} \in S_2, l = 1, 2, \ldots, \#L_2(p)\right\}$ are two PLTSs based on S, where $\#L_1(p) = \#L_2(p) = L$, the generalized mixed distance between them is defined as Equation (12), the parameter $\varsigma \in [0.1]$ and $\lambda \geq 1$.*

$$D_{gh}(L_1(p), L_2(p)) = \left[\varsigma\sum_{l=1}^{L} p^{(l)}\left|\frac{I\left(s_\alpha'^{(l)}\right) - I\left(s_\beta'^{(l)}\right)}{2\tau}\right|^\lambda + (1-\varsigma)\max_{l=1,2,\ldots,L} p^{(l)}\left|\frac{I\left(s_\alpha'^{(l)}\right) - I\left(s_\beta'^{(l)}\right)}{2\tau}\right|^\lambda\right]^{\frac{1}{\lambda}} \quad (7)$$

In the fundamental TOPSIS algorithm, the distance equation, as per Definition 6, is employed. It is essential that two PLTSs, before calculating their distance, possess an identical number of linguistic terms. However, achieving the exact same number of linguistic terms simultaneously for two distinct PLTSs is a challenging task. For example $L_1(p) = \left\{s_\alpha^{(l)}\left(p_\alpha^{(l)}\right) \middle| s_\alpha^{(l)} \in S_1, l = 1, 2, \ldots, \#L_1(p)\right\}$ and $L_2(p) = \left\{s_\beta^{(l)}\left(p_\beta^{(l)}\right) \middle| s_\beta^{(l)} \in S_2, l = 1, 2, \ldots, \#L_2(p)\right\}$ , $\#\#L_1(p) \neq \#L_2(p)$; therefore, two PLTSs with different number of PLTS terms are converted into PLTSs with the same number by Algorithm 1. The $L_1^*(p) = \left\{s_\alpha'^{(l)}\left(p_\alpha^{(l)}\right) \middle| s_\alpha^{(l)} \in S_1, l = 1, 2, \ldots, L\right\}$ and $L_2^*(p) = \left\{s_\beta'^{(l)}\left(p_\beta^{(l)}\right) \middle| s_\beta^{(l)} \in S_2, l = 1, 2, \ldots, L\right\}$.

---

**Algorithm 1.** probability division algorithm [40]:

Input: two PLTS $L_1(p) = \left\{s_\alpha^{(l)}\left(p_\alpha^{(l)}\right) \middle| s_\alpha^{(l)} \in S_1, l = 1, 2, \ldots, L_1\right\}$ and $L_2(p) = \left\{s_\beta^{(l)}\left(p_\beta^{(l)}\right) \middle| s_\beta^{(l)} \in S_2, l = 1, 2, \ldots, L_2\right\}$, the variable *flag* represents the current position in the PLTS, and the variable *sum* stores the probability sum from the PLTS $L_1(p)$ to the *flag*-th linguistic term.

Step 1. Set flag = 1, sum = 0;

Step 2. If $p_\alpha^{(flag)} < p_\beta^{(flag)}$, the $L_2(p)$ of elements in the $s_\beta^{(flag)}\left(p_\beta^{(flag)}\right)$ is divided into two elements, $s_\beta^{(flag)}\left(p_\alpha^{(flag)}\right)$ and $s_\beta^{(flag)}\left(p_\beta^{(flag)} - p_\alpha^{(flag)}\right)$. The former is used to replace the element $s_\beta^{(flag)}\left(p_\beta^{(flag)}\right)$, and the latter is inserted between the flag element and the (flag +1) element in $L_2(p)$;

If $p_\alpha^{(flag)} = p_\beta^{(flag)}$, no operation is performed;

Step 3. $sum = sum + p_\alpha^{(flag)}$;

In Step 4, if the sum is $\geq 1$, proceed to the next step; otherwise, increment the flag by 1 and return to Step 2. The probabilistic splitting algorithm preprocesses the PLTS to ensure a consistent probability distribution. With this algorithm, the design of the generalized mixed weighted distance becomes possible.

---

### 4.2. Post-Occupancy Evaluation Model of Community Comprehensive Elderly Care Service Facilities Based on PLTS

According to the evaluation index system, there are three evaluation attributes, and each evaluation attribute contains a different number of evaluation criteria and sub-criteria. In order to facilitate expression, it is assumed that the number of evaluation criteria under all attributes is $N_c$, while the number of sub-criteria under all criteria is $N_{sc}$. Meanwhile, it is assumed that community comprehensive elderly care service facilities participate in the evaluation. The number of users participating in the evaluation is $N_u$, and the number of experts participating in the evaluation is $N_e$. The evaluation is divided into H evaluation stages. The flow chart of the evaluation model is shown in Figure 1.

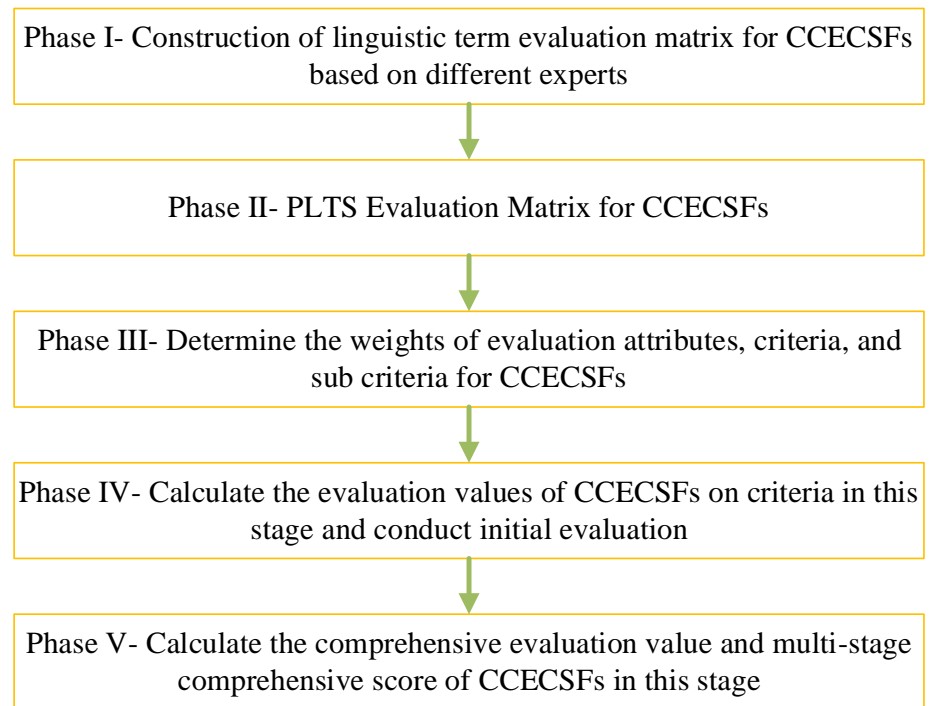

**Figure 1.** Flow chart of the Post-Occupancy evaluation model.

4.2.1. Phase I—Construction of Linguistic Evaluation Matrix of Community Comprehensive Elderly Care Service Facilities Based on Different Experts

The purpose of this phase is to construct the linguistic evaluation matrix for community comprehensive elderly care service facilities, the specific steps of Phase I are shown in Figure 2.

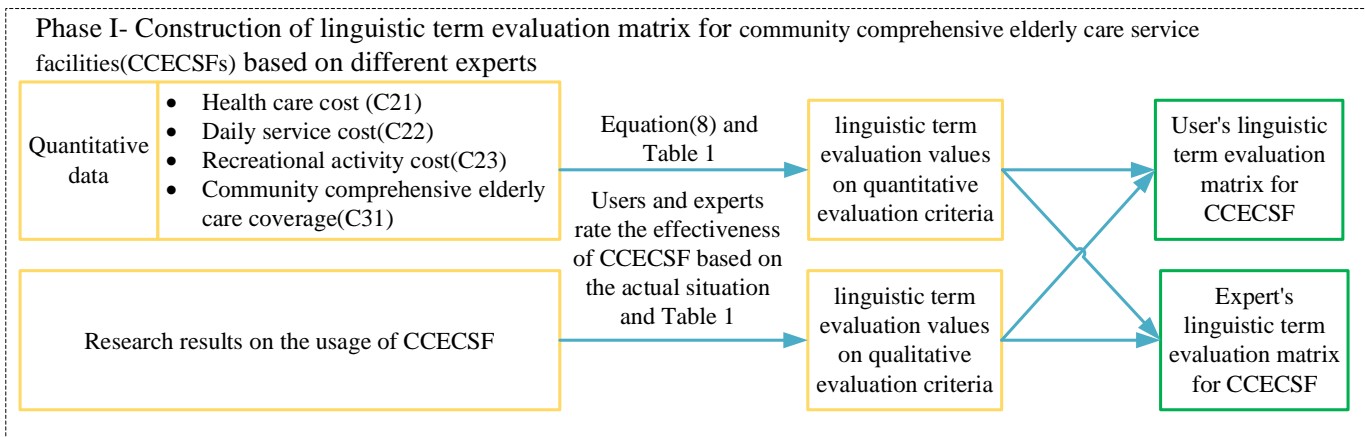

**Figure 2.** Construction of linguistic term evaluation matrix for community comprehensive elderly care service facilities based on different experts.

Step 1: Standardized processing of evaluation data. According to Equation (8) [40], the quantitative evaluation values of the evaluation index system of community comprehensive elderly care service facilities, such as medical care cost (C21), daily service cost (C22), recreational activity cost (C23) and elderly coverage rate (C31) of community comprehensive elderly care service facility, are normalized. Then, the standardized evaluation results are

compared with the evaluation range in Table 1 to determine the linguistic term evaluation value.

$$CL_{ij} = \begin{cases} \frac{EV_{ij} - minEV_i}{maxEV_i - minEV_i}, & i \in \Omega_b \\ \frac{maxEV_i - EV_{ij}}{maxEV_i - minEV_i}, & i \in \Omega_c \end{cases} \tag{8}$$

$EV_i^{max}$ and $EV_i^{min}$ represent the maximum and minimum evaluation values, respectively, based on expert opinions for the $i$-th evaluation criterion. $EV_{ij}$ denotes the evaluation value of the j-th evaluated item concerning the $i$-th evaluation criterion. $\Omega_b$ is the set of positive evaluation criteria, while $\Omega_c$ is the set of negative evaluation criteria.

Step 2: Determine the qualitative evaluation value. According to the qualitative evaluation sub-criterion in Table 2 and the evaluation linguistic term in Table 1, users and experts evaluate the community comprehensive elderly care service facilities to obtain the linguistic evaluation value of the community comprehensive elderly care service facilities in the qualitative evaluation sub-criterion. It should be noted that in the evaluation index system of this paper, qualitative evaluation criteria are all positive criteria. However, if negative evaluation criteria are added in actual use, the negative linguistic evaluation value needs to be converted into positive linguistic evaluation value through Equation (9) [40].

$$\begin{cases} neg(s_\alpha) = s_{-\alpha}, \{S_\alpha | \alpha = -\tau, \dots, -1, 0, -1, \dots, \tau\} \\ neg(s_\alpha) = s_\beta, \beta = 2\tau + 1 - \alpha, \{S_\alpha | \alpha = 1, \dots, 2\tau\} \end{cases} \tag{9}$$

Step 3: Based on the above steps, each expert and each user form their own linguistic evaluation matrix $R^{(k)}$ for the community comprehensive elderly care service facility, as shown in Equation (10).

$$R^{(k)} = \left[ s_{ij}^{(k)} \right]_{m \times n} = \begin{bmatrix} s_{11}^{(k)} & \cdots & s_{1n}^{(k)} \\ \vdots & \ddots & \vdots \\ s_{m1}^{(k)} & \cdots & s_{mn}^{(k)} \end{bmatrix}, \ k = 1, \dots, N_e + N_u \tag{10}$$

where the subscript m is the number of community comprehensive elderly care service facilities, and n is the number of evaluated sub-criterion $N_{sc}$.

### 4.2.2. Phase II—PLTS Evaluation Matrix of Community Comprehensive Elderly Care Service Facilities

The purpose of this phase is to construct the PLTS evaluation matrix for community comprehensive elderly care service facilities, the specific steps of Phase I are shown in Figure 3.

In the evaluation of community comprehensive elderly care service facilities, decision makers can be divided into two categories, namely users of elderly care facilities and experts. Therefore, in this section, the decision weights of users and experts will be determined respectively. Then, the linguistic evaluation matrix given by the two types of decision makers is aggregated into the user group evaluation matrix and the expert group evaluation matrix based on the PLTS. Finally, if a certain criterion needs to be evaluated jointly by users and experts, the decision-making power of users and experts in the evaluation needs to be determined, and the evaluation matrix given by the two types of decision makers is aggregated into the comprehensive evaluation matrix. The specific steps are shown in Figure 3.

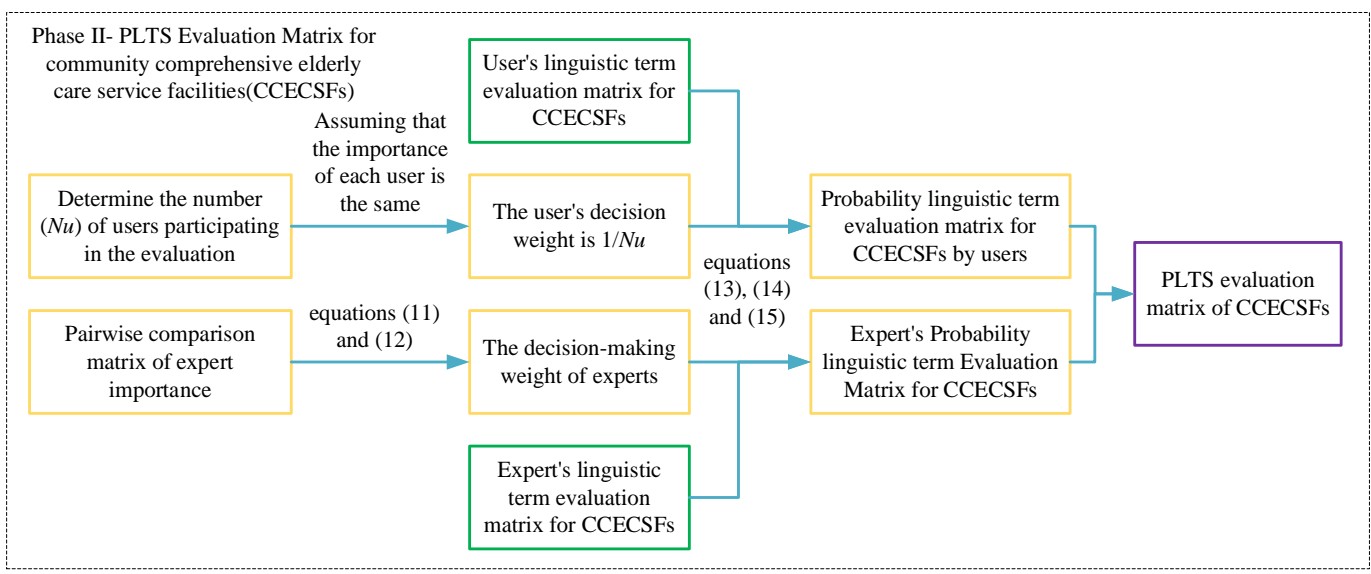

**Figure 3.** PLTS Evaluation Matrix for community comprehensive elderly care service facilities.

Step 1: Weight setting for users of community comprehensive elderly care service facilities. In this paper, users are regarded as individuals with the same decision-making power, so the weight of users is $1/N_u$, where $N_u$ is the number of users;

Step 2: The weight assignment for experts in community comprehensive elderly care service facilities involves a process where the government department assesses the relative importance of experts by conducting pairwise comparisons using PLTS. This comparison results in the creation of a pair comparison matrix: $AR = \left(L_{ij}(p)\right)_{N_e \times N_e}$, where the element $L_{ij}(p)$ represents the relative importance of the ith expert relative to the jth expert. The expert weight $ew_i$ of the *i*-th expert can be obtained by Equations (11) and (12) [40].

$$w'_i = \exp\left(\frac{ln\sqrt{2}}{N_e}\sum_{j=1}^{N_e} S\left(L_{ij}(p)\right)\right); j = 1, 2, \ldots, N_e \tag{11}$$

$$ew_i = w'_i / \sum_{i=1}^{N_e} w'_i \tag{12}$$

Step 3: Aggregate the linguistic evaluation matrices from individual users and experts into group-level evaluation matrices using PLTS. The process involves similar steps for each group, ensuring uniformity in the aggregation method, to express the convenience of users and experts collectively referred to as decision-makers; assume that the weight of decision-makers is $w = \left(w_1, w_2, \ldots, w_{N_d}\right)^T$, where $N_d$ is the number of decision makers, and $N_d$ is determined by the number of users or experts. According to Equations (13)–(15) [40], the community comprehensive elderly care service facility group evaluation matrix can be obtained.

$$R^g = \left[L_{ij}^g(p)\right]_{m \times n} = \begin{bmatrix} L_{11}^g(p) & \cdots & L_{1n}^g(p) \\ \vdots & \ddots & \vdots \\ L_{m1}^g(p) & \cdots & L_{mn}^g(p) \end{bmatrix} \tag{13}$$

where

$$L_{ij}^g(p) = \left\{ s_{ij}^{g(k)}\left(p_{ij}^{g(k)}\right) \middle| s_{ij}^{g(k)} \in \left\{ s_{ij}^{(1)}, \ldots, s_{ij}^{(dn)} \right\}; p_{ij}^{g(k)} = \sum_{q=1}^{N_d} w_q v \right\} \tag{14}$$

$$v = \begin{cases} 1, if\ s_{ij}^{g(k)} = s_{ij}^{(q)} \\ 0, if\ s_{ij}^{g(k)} \notin s_{ij}^{(q)} \end{cases} (q = 1, \ldots, N_d) \tag{15}$$

4.2.3. Phase III—To Determine the Weights of the Evaluation Attributes, Criteria and Sub-Criterion of Community Comprehensive Elderly Care Service Facilities

The weight setting methods of evaluation attributes, evaluation criteria and sub-criteria mainly include subjective weight setting methods, objective weight setting methods, and subjective and objective weight combination methods. Specific weight setting methods are shown in Table 3. The specific steps are shown in Figure 4.

**Table 3.** Setting methods and decision makers for different types of weights.

| Different Types of Weight | Weight Setting Method | Decision Makers |
|---|---|---|
| Evaluation attribute | Subjective weight setting method | Experts |
| Evaluation criterion | Subjective weight setting method | • The criteria below user demand (A1) and user cost (A2) are determined by the user group <br> • Government performance (A3) The criteria below are determined by experts |
| Evaluation sub-criterion | Subjective and objective combination weight setting method | • The criteria below user demand (A1) and user cost (A2) are determined by the user group <br> • Government Performance (A3) The criteria below are determined by experts |

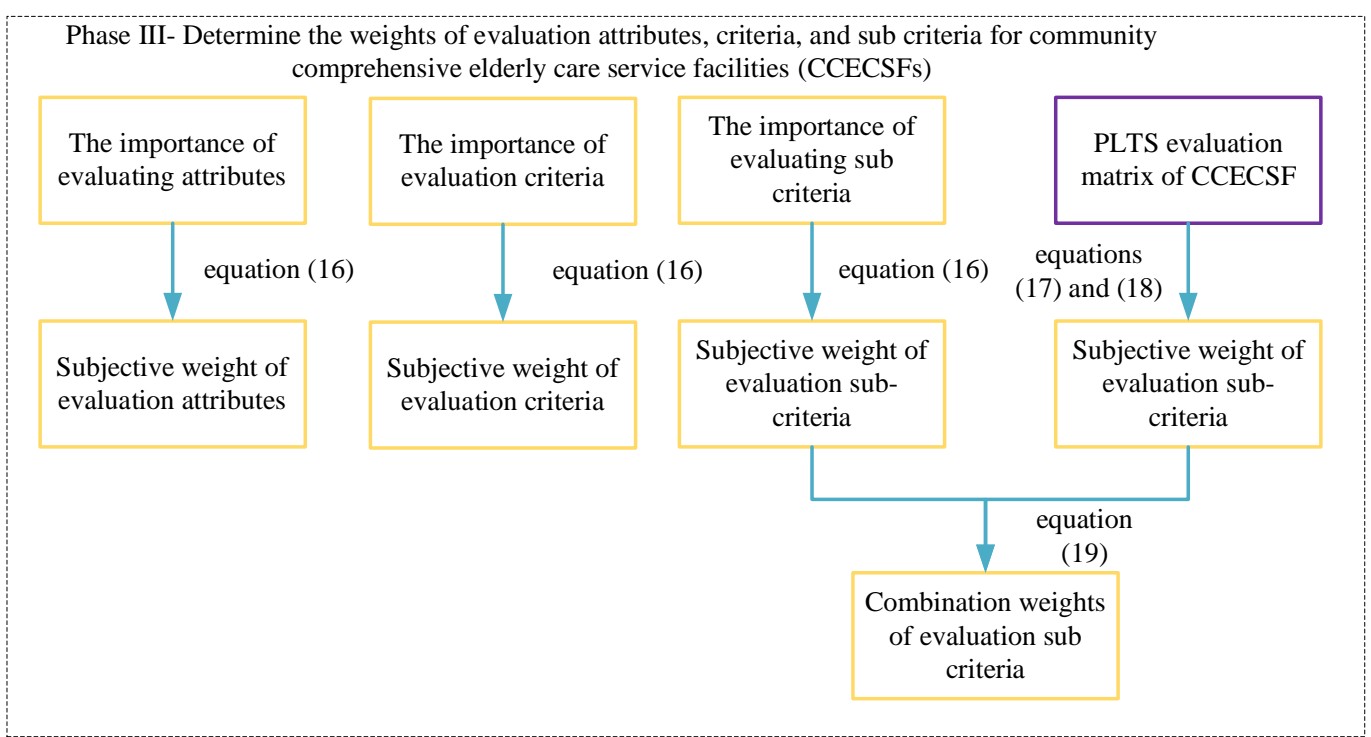

**Figure 4.** Determine the weights of evaluation attributes, criteria and sub criteria for community comprehensive elderly care service facilities.

According to Table 3, the specific setting method is as follows. For the convenience of representing attributes, criteria and sub-criterion, they are collectively referred to as criteria, and there are n criteria.

1. Subjective weight setting method

Both users and experts assign importance degrees to evaluation criteria within the range of [1, 10], where a higher value signifies greater importance. The importance of

the $i$-th evaluation criterion is denoted as $ID_i$. Subsequently, the subjective weight of the evaluation criteria is computed using Equation (16).

$$sw_i = \frac{ID_i}{\sum_{i=1}^{n} ID_i} \tag{16}$$

thereof $\sum_{i=1}^{n} sw_i = 1$.

2.  Objective weight setting method

Firstly, the deviation $dd_{ij}$ of the evaluation value of each PLTS is calculated by Equation (6), then the information entropy on the $i$-th evaluation criterion is calculated by Equation (17), and the objective weight is calculated by Equation (18) [40].

$$En_i = -\frac{1}{ln(n)}\left(\sum_{j=1}^{m} \frac{dd_{ij}}{dd_i^{total}} ln\left(\frac{dd_{ij}}{dd_i^{total}}\right)\right) \tag{17}$$

$$ow_i = \frac{1 - En_i}{\sum_{i=1}^{n}(1 - En_i)} \tag{18}$$

where $dd_{ij}$ is the deviation degree of $L_{ij}(p)$, $dd_i^{total} = \sum_{j=1}^{m} dd_{lj}$, $ow_i$ is the objective weight of the $i$-th criterion, where $\sum_{i=1}^{n} ow_i = 1$.

3.  Subjective and objective combination weights setting method

After determining the subjective and objective weights of the evaluation criteria, the combined weights of the evaluation criteria are calculated by Equation (19) [40].

$$cw_i = \alpha \times sw_i + (1 - \alpha) \times ow_i, \ i = 1, 2, \dots, n \tag{19}$$

where $0 \leq \alpha \leq 1$, $\alpha$ is the combined weight coefficient.

### 4.2.4. Phase IV—Calculate the Evaluation Value of Community Comprehensive Elderly Care Service Facilities (CCECSFs) in the Current Evaluation Stage and Conduct the Initial Evaluation

To aggregate the PLTS evaluation value and transform it into real evaluation value, the PLTS-TOPSIS method is used in this paper to aggregate the PLTS evaluation value on sub-criterion, wherein the positive and negative ideal solutions in TOPSIS method are set after discussion by experts, instead of being selected from the evaluation matrix according to the principle of maximum and minimum. The specific steps of phase IV are shown in Figure 5.

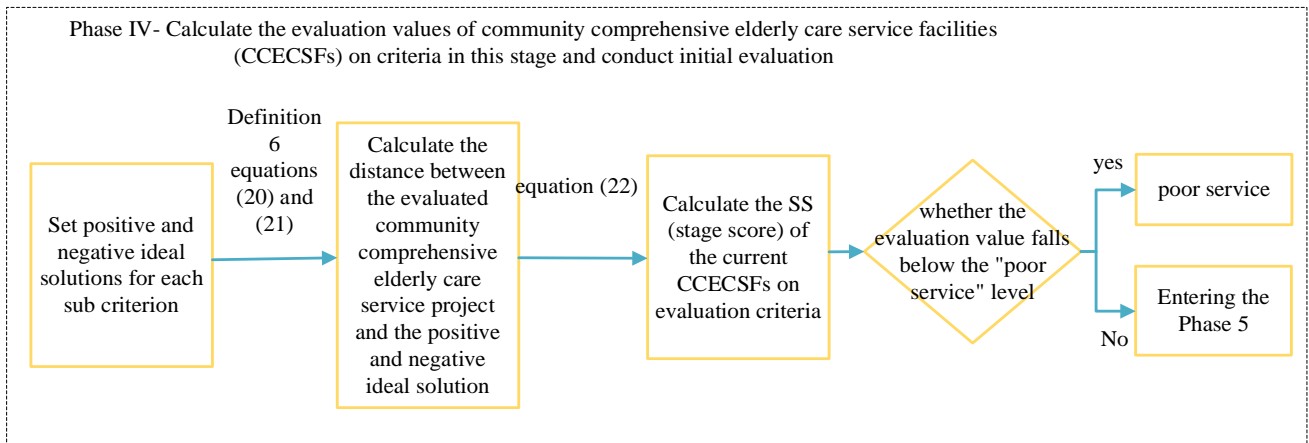

**Figure 5.** Calculate the evaluation values of community comprehensive elderly care service facilities on criteria in this stage and conduct an initial evaluation.

Step 1: Set the positive and negative ideal solutions. According to the expected service level of community comprehensive elderly care service facilities, set the positive ideal solution $L^+ = \left\{ L_1^+(p), \ldots, L_j^+(p), \ldots, L_n^+(p) \right\}$ and the negative ideal solution $L^- = \left\{ L_1^-(p), \ldots, L_j^-(p), \ldots, L_n^-(p) \right\}$, where n is the number of evaluation sub-criteria $N_{sc}$.

Step 2: Calculate the distance between the community comprehensive elderly care service facilities and the positive and negative ideal solutions. According to Definition 6, the distance between the community comprehensive elderly care service facilities and the positive and negative ideal solutions is calculated. For specific equations, see Equations (20) and (21).

$$d(x_j, L^+) = \sum_{i=1}^n d\left(L_{ij}(p), L_i^+(p)\right) cw_{(i)} \tag{20}$$

$$d(x_j, L^-) = \sum_{i=1}^n d\left(L_{ij}(p), L_i^-(p)\right) cw_{(i)} \tag{21}$$

Step 3: Calculate the evaluation stage score (*ess*) of community comprehensive elderly care service facility in the evaluation criterion at the present evaluation stage through Equation (22).

$$ESS_i = \frac{d(x_i, L^-)}{d(x_i, L^-) + d(x_i, L^+)} \tag{22}$$

Step 4: According to Table 4, if the evaluation value on the general demand (C11), user physical demand (C12) and user cognitive demand (C13) falls below the level of "poor service", it will be directly judged as poor service, and the comprehensive evaluation value of the community comprehensive elderly care service facilities takes the lowest score among the above criteria.

**Table 4.** Score range and service level rating.

| Score Range | Service Level Rating |
| --- | --- |
| [0, 0.11] | Horrible service |
| (0.11, 0.22] | Poor service |
| (0.22, 0.33] | Relatively poor service |
| (0.33, 0.66] | General service |
| (0.66, 0.77] | Relatively good service |
| (0.77, 0.88] | Good service |
| (0.88, 1] | Excellent service |

4.2.5. Phase V—Calculate the Comprehensive Evaluation Value and Multi-Stage Comprehensive Scores of Community Comprehensive Elderly Care Service Facilities

Step 1: Calculate the overall score of this evaluation stage. Using the weighted average method, the evaluation value of the criterion is aggregated into the evaluation value of the attributes. On this basis, the evaluation value of the attributes is aggregated into the comprehensive evaluation value by the weighted average method again. The specific steps are shown in Figure 6.

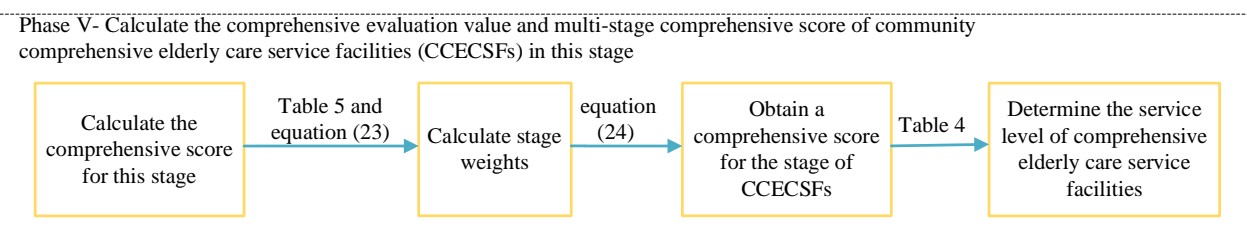

**Figure 6.** Calculate the comprehensive evaluation value and a multi-stage comprehensive score of community comprehensive elderly care service facilities.

Step 2: Calculate the evaluation stage weights. Suppose there are h evaluation stages, and the evaluation stage weights are $\lambda(t) = (\lambda(t_1), \lambda(t_2), \ldots, \lambda(t_h))^T$, $\mu$ are quantified values of experts' preferences for each evaluation stage of the decision object, which can be determined according to Table 5. Finally, the weight of each evaluation stage is calculated according to Equation (23) [36].

$$max\left[-\sum_{\xi=1}^{h}\lambda\left(t_{\xi}\right)ln\lambda\left(t_{\xi}\right)\right]$$

$$s.t.\begin{cases} \mu = \sum_{\xi=1}^{h}\frac{h-\xi}{h-1}\lambda\left(t_{\xi}\right) \\ \sum_{\xi=1}^{h}\lambda\left(t_{\xi}\right) = 1, \lambda\left(t_{\xi}\right) \in [0,1] \end{cases} \tag{23}$$

where $\mu$ indicates the expert's preference for different evaluation stages.

**Table 5.** Quantitative table of experts' phased preferences.

| Evaluation Stage Preference | Implication |
|---|---|
| 0.10 | Attach great importance to the recent events |
| 0.30 | Pay more attention to the latest evaluation stage |
| 0.50 | Give equal attention to the data at each evaluation stage |
| 0.70 | Pay more attention to what happened in the previous evaluation stage |
| 0.90 | Pay attention to what happened in the previous evaluation stage |
| 0.20, 0.40, 0.60, 0.80 | It is used to represent compromised nonharmonic values |

Step 3: Get the comprehensive score of the community comprehensive elderly care service facilities. According to the evaluation stage weights obtained in the previous step, the service level of each evaluation stage of the community comprehensive elderly care service facilities and the Equation (24), the Comprehensive evaluation stage score (CESS) of the community comprehensive elderly care service facilities is calculated. Finally, the service level of the community comprehensive elderly care service facilities is determined according to Table 4.

$$CESS = \sum_{\xi=1}^{h} ESS_{\xi}\lambda\left(t_{\xi}\right) \tag{24}$$

## 5. Case Study

In this section, an example is given to prove the validity of the model proposed in this paper. In Jinan City, Shandong Province, China, three community comprehensive elderly care service facilities under the PPP mode were selected for post-occupancy evaluation. These three community comprehensive elderly care service facilities are marked as NH1, NH2 and NH3, among which NH1 is a newly built community comprehensive elderly care service facilities facility, NH2 is an old community comprehensive elderly care service facilities facility, but it has been renovated and later rebuilt, and NH3 is an old community comprehensive elderly care service facility. The number of people served by the three elderly care facilities is as follows: NH1 is 202 people, NH2 is 151 people and NH3 is 73 people.

To effectively evaluate political performance, three experts are selected. When selecting three experts, it is crucial to consider their expertise, experience and reputation in relevant fields to ensure the accuracy and scientific validity of political performance evaluations, and the average work experience of experts is about 3 years. For the political performance evaluation discussed in this article, here are the reasons for selecting three experts:

Government Management Expert: This expert should possess extensive experience in government management and a profound understanding of the PPP model. Since political performance evaluation is closely related to the government's role in PPP projects, they can provide insights into how the government influences project success. Additionally, they can assess the efficiency and effectiveness of government management and operation in

eldercare service facilities. Such an expert can help ensure that the government achieves the expected performance level in PPP projects.

Elderly Care Industry Expert: This expert should have rich experience in the elderly care industry and a deep understanding of the operation and management of elderly care facilities. They can evaluate the operational aspects of elderly care facilities, including service quality, scope of services and satisfaction levels. Given that the selected elderly care facilities are community-based comprehensive care facilities, this expert can provide insights into the best practices and industry trends in community-based elderly care services.

PPP mode Expert: This expert should be proficient in PPP project management and possess expertise in PPP contract structures, risk allocation and collaboration models. They can evaluate the application effectiveness of the PPP mode in managing elderly care facilities and offer insights into the best practices and experiences of the PPP mode in the elderly care sector. Since the selected facilities operate under the PPP mode, this expert can provide professional insights into the PPP mode's application in the elderly care sector.

Selecting the aforementioned three experts ensures an effective evaluation of political performance and provides valuable recommendations for the subsequent improvement of elderly care facilities. These three experts will be able to provide comprehensive analysis and suggestions for political performance evaluation from different perspectives and professional fields.

According to Table 6, the linguistically important pairwise comparison matrix among the three experts is determined, and then the weight of the experts is calculated according to Equations (11) and (12). The comparison matrix and expert weights are shown in Table 7. The experts evaluated the community comprehensive elderly care service facilities according to the actual situation, and the evaluation results are shown in Table 8. Finally, according to Equations (13)–(15), the PLTS evaluation values of sub-criterion C321 and C322 were obtained, and the details are shown in Table 9.

As for the criteria under the user demand attribute, the users of community comprehensive elderly care facilities evaluate NH1, NH2 and NH3 according to their own experience and Table 1, and obtain the evaluation value based on PLTS under the user demand attribute through Equations (13)–(15), as shown in Table 9 for details. What needs to be explained here is: Users in NH1 have a weight of 1/202, NH2 users have a weight of 1/151, and NH3 users have a weight of 1/73.

**Table 6.** Linguistic terms for important comparison.

| Symbols | Description |
| --- | --- |
| $S_{-4}$ | absolutely less important |
| $S_{-3}$ | much less important |
| $S_{-2}$ | slightly less important |
| $S_{-1}$ | less important |
| $S_0$ | equally important |
| $S_1$ | slightly more important |
| $S_2$ | more important |
| $S_3$ | much more important |
| $S_4$ | absolutely more important |

**Table 7.** Expert importance pair-to-pair comparison matrix and expert weight.

| | Expert1 | Expert2 | Expert3 | Weights |
| --- | --- | --- | --- | --- |
| Expert1 | $S_0$ | $S_{-1}$ | $S_1$ | 0.32 |
| Expert2 | $S_1$ | $S_0$ | $S_2$ | 0.45 |
| Expert3 | $S_{-1}$ | $S_{-2}$ | $S_0$ | 0.23 |

**Table 8.** Experts' evaluation of community comprehensive elderly care service facility on criterion C32.

| | Expert1 | | | Expert2 | | | Expert3 | | |
| | NH1 | NH2 | NH3 | NH1 | NH2 | NH3 | NH1 | NH2 | NH3 |
|---|---|---|---|---|---|---|---|---|---|
| C321 | $S_3$ | $S_2$ | $S_1$ | $S_3$ | $S_2$ | $S_2$ | $S_2$ | $S_2$ | $S_1$ |
| C322 | $S_3$ | $S_3$ | $S_1$ | $S_3$ | $S_2$ | $S_2$ | $S_3$ | $S_2$ | $S_2$ |

**Table 9.** Post-occupancy evaluation matrix of community comprehensive elderly care service facility based on PLTS.

| | NH1 | NH2 | NH3 |
|---|---|---|---|
| C111 | $\{S_1(0.33), S_2(0.37), S_30.3)\}$ | $\{S_1(0.32), S_2(0.4), S_3(0.28)\}$ | $\{S_1(0.32), S_2(0.38), S_3(0.3)\}$ |
| C112 | $\{S_1(0.31), S_2(0.36), S_3(0.33)\}$ | $\{S_1(0.3), S_2(0.35), S_3(0.34)\}$ | $\{S_1(0.33), S_2(0.38), S_3(0.29)\}$ |
| C113 | $\{S_1(0.29), S_2(0.35), S_3(0.36)\}$ | $\{S_1(0.32), S_2(0.32), S_3(0.36)\}$ | $\{S_1(0.32), S_2(0.32), S_3(0.37)\}$ |
| C114 | $\{S_1(0.29), S_2(0.42), S_3(0.29)\}$ | $\{S_1(0.3), S_2(0.39), S_3(0.3)\}$ | $\{S_1(0.29), S_2(0.4), S_3(0.32)\}$ |
| C121 | $\{S_1(0.3), S_2(0.39), S_3(0.31)\}$ | $\{S_1(0.31), S_2(0.38), S_3(0.3)\}$ | $\{S_1(0.29), S_2(0.4), S_3(0.32)\}$ |
| C122 | $\{S_1(0.32), S_2(0.35), S_3(0.34)\}$ | $\{S_1(0.28), S_2(0.36), S_3(0.35)\}$ | $\{S_1(0.27), S_2(0.42), S_3(0.3)\}$ |
| C123 | $\{S_1(0.31), S_2(0.4), S_3(0.3)\}$ | $\{S_1(0.3), S_2(0.39), S_3(0.3)\}$ | $\{S_1(0.29), S_2(0.4), S_3(0.32)\}$ |
| C131 | $\{S_1(0.32), S_2(0.31), S_3(0.37)\}$ | $\{S_1(0.32), S_2(0.28), S_3(0.39)\}$ | $\{S_1(0.38), S_2(0.3), S_3(0.32)\}$ |
| C132 | $\{S_0(0.3), S_1(0.38), S_2(0.32)\}$ | $\{S_0(0.27), S_1(0.39), S_2(0.34)\}$ | $\{S_0(0.22), S_1(0.41), S_2(0.37)\}$ |
| C211 | $\{S_3(1)\}$ | $\{S_3(1)\}$ | $\{S_4(1)\}$ |
| C221 | $\{S_3(1)\}$ | $\{S_3(1)\}$ | $\{S_4(1)\}$ |
| C231 | $\{S_3(1)\}$ | $\{S_3(1)\}$ | $\{S_4(1)\}$ |
| C311 | $\{S_3(1)\}$ | $\{S_2(1)\}$ | $\{S_{-1}(1)\}$ |
| C321 | $\{S_2(0.23), S_3(0.77)\}$ | $\{S_2(1)\}$ | $\{S_1(0.55), S_2(0.45)\}$ |
| C322 | $\{S_3(1)\}$ | $\{S_2(0.68), S_3(0.32)\}$ | $\{S_1(0.32), S_2(0.68)\}$ |

In the post-occupancy index system, the criterion in the user cost attribute is quantitative, so the data in the market are collected according to the criterion, and Table 10 is obtained. The corresponding evaluation value based on PLTS is obtained by Equation (8). See Table 9 for details, in which the maximum and minimum values of each criterion are determined by experts according to the market situation.

**Table 10.** Quantitative evaluation value of community comprehensive elderly care service facility on user cost attribute.

| | NH1 | NH2 | NH3 | Max | Min |
|---|---|---|---|---|---|
| C211 | 400 | 400 | 350 | 600 | 350 |
| C221 | 600 | 550 | 500 | 700 | 500 |
| C231 | 100 | 100 | 70 | 200 | 60 |
| C311 | 95 | 90 | 75 | 100 | 70 |

Then, users and experts score the importance of each attribute, criteria and sub-criteria according to their own importance and responsibility (see Table 3 for details). After scoring, the subjective weight of the attribute, criteria and sub-criteria is calculated by Equation (16), as shown in Tables 11–14. Then, based on Table 9 and Equations (17) and (18), the objective weights and combined weights of the sub-criteria are calculated, as shown in Tables 11 and 12.

**Table 11.** Subjective weights, objective weights and combined weights of sub-criteria under user demand and cost attributes.

|  | Average Scores | Subjective Weights | Objective Weights | Combination Weight |
|---|---|---|---|---|
| C111 | 7.99 | 0.24 | 0.25 | 0.25 |
| C112 | 7.10 | 0.21 | 0.25 | 0.23 |
| C113 | 7.97 | 0.24 | 0.25 | 0.25 |
| C114 | 10.00 | 0.31 | 0.25 | 0.28 |
| C121 | 8.91 | 0.33 | 0.01 | 0.17 |
| C122 | 8.99 | 0.33 | 0.99 | 0.66 |
| C123 | 9.00 | 0.33 | 0 | 0.17 |
| C131 | 8.99 | 0.51 | 0.5 | 0.51 |
| C132 | 8.50 | 0.49 | 0.5 | 0.50 |
| C211 |  | 1 | 1 | 1.00 |
| C221 |  | 1 | 1 | 1.00 |
| C231 |  | 1 | 1 | 1.00 |
| C311 |  | 1 | 1 | 1.00 |

**Table 12.** The subjective weight, objective weight and combined weight of sub-criteria under the political performance attribute.

|  | Expert1 | Expert2 | Expert3 | Group | Subjective Weights | Objective Weights | Combination Weight |
|---|---|---|---|---|---|---|---|
| C321 | 9 | 8 | 8 | 8.32 | 0.49 | 1 | 0.75 |
| C322 | 9 | 9 | 8 | 8.77 | 0.51 | 0 | 0.26 |

**Table 13.** Criteria weights.

|  | Average Scores | Subjective Weights |
|---|---|---|
| C11 | 8.03 | 0.31 |
| C12 | 9.04 | 0.35 |
| C13 | 9.01 | 0.34 |
| C21 | 9.00 | 0.36 |
| C22 | 9.00 | 0.36 |
| C23 | 7.03 | 0.28 |
| C31 | 9 | 0.5 |
| C32 | 9 | 0.5 |

**Table 14.** Attribute weights.

|  | Average Scores | Subjective Weights |
|---|---|---|
| A1 | 9 | 0.38 |
| A2 | 7.67 | 0.32 |
| A3 | 7.33 | 0.30 |

Based on Tables 9 and 11, the positive ideal solution and the negative ideal solution are set as $\{S_4(1)\}$ and $\{S_{-4}(1)\}$, respectively. According to Equations (20) and (21), the distances between the evaluation value of NH1, NH2 and NH3 on the sub-criteria and the positive and negative ideal solutions were calculated, as shown in Table 15. On this basis, through Equation (22), the evaluation values on the criteria were calculated, and the results are shown in Table 16. Through this step, the PLTS evaluation values were aggregated into the evaluation value based on real numbers, thus simplifying the subsequent calculation complexity. After completing this step, the initial evaluation found that NH3 had the lowest value on C13, 0.68, which belonged to the grade of "good service" according to Table 4, so NH1, NH2 and NH3 were all above the pass line.

**Table 15.** Distances to positive and negative ideal solutions.

| | The Weighted Distance to the Positive Ideal Solution | | | The Weighted Distance to the Negative Ideal Solution | | |
|---|---|---|---|---|---|---|
| | NH1 | NH2 | NH3 | NH1 | NH2 | NH3 |
| C11 | 0.18 | 0.18 | 0.18 | 0.53 | 0.53 | 0.52 |
| C12 | 0.18 | 0.18 | 0.18 | 0.52 | 0.53 | 0.53 |
| C13 | 0.22 | 0.22 | 0.23 | 0.49 | 0.50 | 0.49 |
| C21 | 0.13 | 0.13 | 0.00 | 0.88 | 0.88 | 1.00 |
| C22 | 0.13 | 0.13 | 0.00 | 0.88 | 0.88 | 1.00 |
| C23 | 0.13 | 0.13 | 0.00 | 0.88 | 0.88 | 1.00 |
| C31 | 0.13 | 0.25 | 0.63 | 0.88 | 0.75 | 0.38 |
| C32 | 0.13 | 0.23 | 0.25 | 0.79 | 0.72 | 0.54 |

**Table 16.** Composite scores based on criteria.

| Criteria | NH1 | NH2 | NH3 |
|---|---|---|---|
| C11 | 0.74 | 0.74 | 0.74 |
| C12 | 0.74 | 0.75 | 0.75 |
| C13 | 0.69 | 0.70 | 0.68 |
| C21 | 0.88 | 0.88 | 1.00 |
| C22 | 0.88 | 0.88 | 1.00 |
| C23 | 0.88 | 0.88 | 1.00 |
| C31 | 0.88 | 0.75 | 0.38 |
| C32 | 0.86 | 0.76 | 0.68 |

Then, according to the weighted average method, the comprehensive evaluation values of NH1, NH2 and NH3 in attributes and the current evaluation stage was obtained, as shown in Table 17. The comprehensive evaluation value of NH1, NH2 and NH3 in the previous evaluation stages is shown in Table 18. According to Table 5, experts believe that more attention should be paid to the most recent evaluation stage, so $\mu = 0.3$ is selected. According to Equation (23), the weight of each evaluation stage is calculated in Table 19, and the final multi-stage evaluation results are shown in Table 18. NH2 and NH3 have the same post-occupancy.

**Table 17.** Composite scores and final results based on attributes.

| | NH1 | NH2 | NH3 |
|---|---|---|---|
| A1 | 0.72 | 0.73 | 0.72 |
| A2 | 0.88 | 0.88 | 1.00 |
| A3 | 0.87 | 0.75 | 0.53 |
| Final | 0.82 | 0.78 | 0.75 |

**Table 18.** The scores of the first four post-occupancy evaluations of the participating community comprehensive elderly care service facility.

| | NH1 | NH2 | NH3 |
|---|---|---|---|
| First evaluation stage | 0.69 | 0.69 | 0.78 |
| Second evaluation stage | 0.70 | 0.65 | 0.76 |
| Third evaluation stage | 0.75 | 0.72 | 0.73 |
| Fourth evaluation stage | 0.77 | 0.74 | 0.74 |
| Fifth evaluation stage | 0.82 | 0.78 | 0.75 |
| Final | 0.77 | 0.74 | 0.74 |

**Table 19.** Evaluation stage weights of the five evaluation stages of low-carbon development level.

|  | First Evaluation Stage | Second Evaluation Stage | Third Evaluation Stage | Fourth Evaluation Stage | Fifth Evaluation Stage |
|---|---|---|---|---|---|
| Stage weights | 0.00 | 0.15 | 0.23 | 0.29 | 0.33 |

According to Tables 16–18, it can be found that NH2 performs the best in terms of user demand because the facilities there are reconstructed based on the original users' opinions, so they can better reflect the needs of users. The three facilities have the same score in terms of the general needs of users, but there were slight differences in physical needs (C12) and cognitive needs (C13). In terms of user cost, NH3 is better because it makes the old facilities charge less. In terms of political performance, NH3 is the least effective because it cannot accommodate more people, while NH1 is the best because it is the largest newly built retirement facility in the region.

## 6. Discussion

To assess the robustness of the post-occupancy value in the current evaluation stage, weights are represented by real numbers, which themselves have a poor ability to handle uncertainty; meanwhile, the evaluation values are represented by PLTS, which can handle uncertainty and does not require sensitivity analysis through floating evaluation values. For the abovementioned reason, there are two types of sensitivity analyses for the weights that were conducted.

The first involved adjusting the parameter "a" in Equation (19), with a value range of [0, 1]. This yielded a total of 11 sensitivity analysis results, detailed in Table 20. The computational findings indicated stable evaluation results. The second type of analysis focused on altering attribute weights, ranging from a 30% reduction to a 30% increase, resulting in seven outcomes presented in Table 21. To demonstrate the superiority of the proposed model in this paper, a comparative analysis was conducted, divided into two cases, the detail could be seen in Table 22. The first case excluded the consideration of PLTS probability and TOPSIS aggregation principle, leading to ranking results with limited changes but increased difficulty in interpretation, requiring manual judgment. This introduced randomness to the post-occupancy results. In the second scenario, focusing solely on the post-occupancy comprehensive value of the current evaluation stage, NH1 and NH2 achieved a "good service" level, while NH3 remained at the same level. This change indicated an improvement in the post-occupancy evaluation level of NH2. However, this improvement was attributed to the limitation of solely considering the post-occupancy score, which could not reflect the sustainability of the use effect. Table 18 illustrates that NH2 consistently ranked last in terms of use effect among the three in the past. Notably, after the completion of the fifth evaluation stage of transformation, NH2's use effect experienced a significant improvement.

**Table 20.** Sensitivity analysis of the first category.

| a | 0.00 | 0.10 | 0.20 | 0.30 | 0.40 | 0.50 |
|---|---|---|---|---|---|---|
| S1 | 0.81 | 0.81 | 0.81 | 0.81 | 0.81 | 0.81 |
| S2 | 0.78 | 0.78 | 0.78 | 0.78 | 0.78 | 0.78 |
| S3 | 0.75 | 0.75 | 0.75 | 0.75 | 0.75 | 0.75 |
| **a** | **0.60** | **0.70** | **0.80** | **0.90** | **1.00** | |
| S1 | 0.82 | 0.82 | 0.82 | 0.82 | 0.82 | |
| S2 | 0.78 | 0.78 | 0.78 | 0.78 | 0.78 | |
| S3 | 0.75 | 0.75 | 0.75 | 0.75 | 0.75 | |

**Table 21.** Sensitivity analysis of the second category.

|  | −30% | −20% | −10% | 0 |
|---|---|---|---|---|
| Attribute weight 1 | [0.83, 0.79, 0.76] | [0.83, 0.79, 0.75] | [0.82, 0.79, 0.75] | [0.81, 0.78, 0.75] |
| attribute weight 2 | [0.81, 0.77, 0.72] | [0.81, 0.77, 0.73] | [0.81, 0.78, 0.74] | [0.81, 0.78, 0.75] |
| attribute weight 3 | [0.81, 0.79, 0.78] | [0.81, 0.79, 0.77] | [0.81, 0.78, 0.76] | [0.81, 0.78, 0.75] |
|  | **10%** | **20%** | **30%** | |
| attribute weight 1 | [0.81, 0.78, 0.75] | [0.80, 0.78, 0.75] | [0.80, 0.77, 0.75] | |
| attribute weight 2 | [0.82, 0.79, 0.76] | [0.82, 0.79, 0.78] | [0.82, 0.80, 0.79] | |
| attribute weight 3 | [0.82, 0.78, 0.74] | [0.82, 0.78, 0.73] | [0.82, 0.78, 0.72] | |

**Table 22.** Comparative analysis.

| PLTS | Method | | Rank | | |
|---|---|---|---|---|---|
|  |  |  | **S1** | **S2** | **S3** |
| Regardless of probability | • Calculate the comprehensive assessment value according to the principles in Definition 1; <br> • Expert weight, attribute weight and criteria weight remain unchanged; <br> • The weighted sum method is adopted | Symbol | $s_{2.54}$ | $s_{2.29}$ | $s_{2.06}$ |
|  |  | Rank | 1 | 2 | 3 |
| Consider only the present evaluation stage | The evaluation results at this evaluation stage are the final results | Score | 0.82 | 0.78 | 0.75 |

The sensitivity analysis demonstrates the robustness of the evaluation model presented in this paper. The comparative analysis reveals that employing PLTS for data representation of evaluation information and utilizing TOPSIS as the data aggregation model enables the model to effectively handle uncertainty in evaluation information and streamline the aggregation of extensive data sets. Moreover, the outcomes of multi-stage comprehensive evaluations aptly capture the sustainability of the use effect. These measures collectively uphold the scientific integrity of the evaluation results.

## 7. Conclusions

To ensure optimal and sustainable services from community comprehensive elderly care service facilities under the PPP mode, a systematic evaluation of their usage effectiveness is essential. However, the current research has the following issues: first, according to the literature review of post-occupancy evaluation index system of community comprehensive elderly care service facilities (Section 2.1), there is a lack of a comprehensive post-occupancy evaluation index system for community comprehensive elderly care service facilities under the PPP mode; according to the literature review of post-occupancy evaluation models of community comprehensive elderly care service facilities (Section 2.2), the post-occupancy evaluation model faces the problems of uncertainty, aggregation of decision-making preferences among large groups, and sustained evaluation of evaluation results.

Building upon prior research on post-occupancy evaluation of community comprehensive elderly care service facilities, this paper establishes a comprehensive evaluation index system for community comprehensive elderly care service facilities under the PPP mode. The system encompasses three evaluation attributes, eight criteria and fifteen sub-criteria. These attributes focus on user demand, user cost and political performance. This framework aims to offer a scientifically guided approach for the post-occupancy evaluation of community comprehensive elderly care service facilities under the PPP mode.

Building on this foundation, PLTS is employed to address uncertainties in post-occupancy evaluation information and aggregate opinions from large groups. This enhances the robustness of evaluation results. A multi-stage evaluation model is adopted to ensure sustainability in assessing usage effectiveness. In this model, the TOPSIS method, with fixed positive and negative ideal solutions, is used to aggregate evaluation information at each stage. This approach offers advantages such as vertical comparison of low-carbon

development levels and simplifying subsequent calculations while retaining decision information. The paper also incorporates the information entropy method, grounded in expert evaluation stage preferences, to determine stage weights objectively. This ensures objectivity in stage weights while accommodating subjective preferences. Leveraging the evaluation index system and multi-stage model, the paper constructs a post-occupancy evaluation framework for community comprehensive elderly care service facilities under the PPP mode, aiming for a scientifically grounded evaluation.

The economic and commercial impact, teaching impact and policy impact of this study on community comprehensive elderly care service facilities under PPP mode are as follows:

(1) Practical guidance: By establishing a post-evaluation framework, the study provides practical guidance to enable government departments, senior care providers and private investors to better evaluate and improve the operation and effectiveness of community-based integrated senior care facilities under the PPP mode to meet the needs of older people.

(2) Promoting sustainable development: The introduction of the PPP mode will incorporate social capital into the field of elderly care services, which will help ease the pressure on government finances and promote the sustainable development of the elderly care service industry. The post-evaluation framework provided in this study can help optimize resource allocation and improve service efficiency to achieve longer-term sustainable development goals.

(3) Improve service quality: Through the scientific evaluation framework, the operation of community comprehensive elderly care service facilities can be more accurately evaluated, problems can be found and improvement measures can be proposed to improve service quality and improve the quality of life of the elderly.

(4) Promoting the innovation of the elderly care model: This study not only focuses on empirical research, but also pays attention to the scientific and methodological evaluation framework, which helps to promote the innovation and progress of the elderly care service model. The post-evaluation research of community comprehensive elderly care service facilities under the PPP mode can provide experience and reference for the design and improvement of future elderly care service models.

This study delineates two critical avenues for future exploration. Firstly, there exists an imperative necessity to meticulously delineate and quantify qualitative evaluation criteria, rendering them amenable to data-driven decision-making within the realm of big data analytics. Secondly, the current investigation overlooks the interrelation among evaluation criteria, a facet with substantive implications for the scientific rigor of the evaluation process. Consequently, delving into methodologies for the systematic quantification of correlations among post-occupancy evaluation criteria for community comprehensive elderly care service facilities under the Public-Private Partnership (PPP) model emerges as a pivotal focus for further scholarly inquiry.

**Author Contributions:** Conceptualization, Y.T.; Methodology, S.G.; Software, L.L.; Data curation, L.Z.; Writing—original draft, Y.Z. All authors have read and agreed to the published version of the manuscript.

**Funding:** Natural Science Foundation of Shandong Province (grant number ZR2021MG054, ZR2021MG050); The Shandong Province Social Science Foundation of China (grant number 21CGLJ26); The Shandong Jianzhu university doctoral foundation project (grant number X20001Z, X19009S).

**Data Availability Statement:** The original contributions presented in the study are included in the article, further inquiries can be directed to the corresponding authors.

**Conflicts of Interest:** The authors declare no conflict of interest.

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
