# Peer review of "Research on Multi-Stage Post-Occupancy Evaluation Framework of Community Comprehensive Elderly Care Service Facilities under the Public-Private Partnership Mode—A Case Study of China"

_buildings, doi:10.3390/buildings14051343_

Round 1

Reviewer 1 Report

Comments and Suggestions for Authors

1. Abstract: The paper lacks clarity in explaining the novelty of the proposed framework and its potential impact on the field. It would be beneficial to explicitly state the unique contributions of the research and its implications for the industry and society. Keyword should not be too long like community comprehensive elderly care service facility (CCECSF), abbreviation should be avoided.

  1. There are a few instances where significant works seem to be overlooked. For example, there is no mention of specific studies or frameworks related to post-evaluation of elderly care services in other places with similar demographic challenges. Exploring the Market Requirements for Smart and Traditional Ageing Housing Units: A Mixed Methods Approach. Smart Cities 2022, 5, 1752-1775. It would strengthen the paper to acknowledge and discuss relevant international studies and their potential applicability to the Chinese context.
  1. The integration of Chinese culture and lifestyle habits in constructing the evaluation index system is a commendable approach. However, the methodology section lacks sufficient detail regarding the research design and data collection methods. It would be beneficial to provide more information on the specific steps taken to develop the evaluation index system and the rationale behind the selection of the probabilistic linguistic term set and large group decision-making model. The author should state more what are the other index methods
    Predicting Carpark Prices Indices in Hong Kong Using AutoML, Computer Modeling in Engineering & Sciences 2023, 134(3), 2247-2282 to state why yours are better one. Additionally, the paper does not address potential limitations or challenges encountered during the research process. Including a discussion of these aspects would enhance the transparency and rigor of the methodology.
  1. The results are presented clearly, but there is room for improvement in the analysis and interpretation of the findings. The paper should provide a more detailed discussion of the specific outcomes and their implications for evaluating CCECSF. Such as the comprehensive evaluation value is PLTS {𝑆−2(0.4), 𝑆−1(0.2), 𝑆0(0.1), 𝑆0(0.1),𝑆1(0.1),𝑆2(0.2)}, while the scoring standard set of CCECSFs is [𝑆−2(𝑣𝑒𝑟𝑦 𝑝𝑜𝑜𝑟), 𝑆−1(𝑝𝑜𝑜𝑟), 𝑆0(𝑚𝑒𝑑𝑖𝑢𝑚),𝑆1(𝑔𝑜𝑜𝑑) ,𝑆2(𝑣𝑒𝑟𝑦 𝑔𝑜𝑜𝑑)], what are these? Definitions need to state what it meant for ALL symbols. The analysis could benefit from additional statistical or qualitative evidence to support the conclusions drawn. Furthermore, the paper does not sufficiently tie the results to the other elements of the paper, such as the methodology and literature review. The conclusions should be more explicitly linked to the research question and the overall framework proposed in the paper.
  1. The paper partially identifies implications for research, practice, and society. It recognizes the need to bridge the gap between theory and practice in evaluating CCECSF and suggests that the proposed framework can contribute to improving the quality of care for the elderly in the community. However, the discussion on the economic and commercial impact, teaching implications, and policy influence is limited. The paper could provide more specific examples and recommendations on how the research findings can be practically applied and integrated into policy-making and service delivery. Aligning the implications with the findings and conclusions of the paper would strengthen the practical significance of the research.
  1. Quality of Communication: There are grammatical errors. Copy edit is necessary to ensure the paper meets the expected standards of the journal's readership. Additionally, the paper could benefit from clearer and more concise writing, especially in the introduction and discussion sections. Avoiding abbrevation would also improve the overall readability.

Other specific comments:

  • The research question is not explicitly stated in the introduction. It would be helpful to clearly state the research question or objective to guide the readers' understanding of the paper.
  • The interpretations and conclusions need stronger support from the evidence presented. Ensure that the assumptions, methodology, and evidence are logically connected to the conclusions drawn.
  • Many parts are very difficult to understand, for example in discussion: The first involved adjusting the param- eter "a" in equation (25), with a value range of [0,1]. This yielded a total of 11 sensitivity?
    analysis results,
  • Too many abbreviations like CCECSF make it very difficult to read.
  • All parts of the text, references, graphics, and tables should be necessary and contribute to the understanding of the new results and main points. Review the manuscript to eliminate any extraneous information that does not directly support the research findings.
  • The conclusions and potential impacts of the paper need to be clearer. Specifically, provide a concise summary of the key findings and their implications for research, practice, and society. Ensure that the conclusions align with the evidence presented in the paper.
  • The abstract provides a summary of the paper, but it could be more concise and focused. Consider revising the abstract to highlight the main results and their significance, originality, resarch gaps that have been filled by the research.

Comments on the Quality of English Language

Polish English

Author Response

Dear reviewer:

We are very grateful to your comments for the manuscript. According with your advice, we tried our best to amend the relevant part and made some changes in the manuscript. All your questions have been answered in detail in the attachment.

我们衷心感谢审稿人的热情工作,并希望更正能得到认可。如果您有任何问题,请随时与我们联系。

再次非常感谢您的意见和建议。

此致

赵玉林

Reviewer 2 Report

Comments and Suggestions for Authors

This paper is a case study which provides a comprehensive models. Below are my comments for improving the manuscript.

1) Conclusion section must be summarized.

2) Providing a comparison table and summary table is appreciated.

3) the novelty idea of work must be clarified in the introduction section.

Author Response

Dear reviewer:

We are very grateful to your comments for the manuscript. According with your advice, we tried our best to amend the relevant part and made some changes in the manuscript. All your questions have been answered in detail in the attachment

We appreciate for Reviewers’ warm work earnestly, and hope that the correction will meet with approval. Should you have any questions, please contact us without hesitate. 

Once again, thank you very much for your comments and suggestions.

Yours Sincerely,

Zhao Yulin

Reviewer 3 Report

Comments and Suggestions for Authors

1. Introduction, para. 3 - author may consider to rephrase their arguments and criticism on the mentioned-authors (Baldwin and Richard) findings ..since they said "...research results is not strong". I found this to be a harsh statement, I do understand if the finding is not suitable to the China's context, but ethically it is not a good practice saying the author's result as "not strong". Argumentation is good, but consider to rephrase it in a much more appropriate sentence structure.

Heading 2.3 Motivation and innovation of this paper: The author is suggested to remove this section. Justification of study can be combined into the intro section (or at the end of introduction) and not requires to have a specific section in this paper.   

Heading 3.2, line 307-309 - please mention the value or number of users (Nu) and experts (Ne) involved in this POE study. 

Table 5 and Table 7 - the table captions and content is similar. Is this a redundant table? Table 5 explains on the score range for general demand, physical demand and cognitive demand. Table 7 explains on the serve level based on CCESF. It is quite confusing as the content is similar! Please revise the table captions if it is based on different criterion or remove one of the table.

Heading 4, line 462 - On what basis does the three experts were selected? Please provide justification or inclusion criteria of the experts. Why they are chosen?

Author Response

Dear reviewer:

We are very grateful to your comments for the manuscript. According with your advice, we tried our best to amend the relevant part and made some changes in the manuscript. All your questions have been answered in detail in the attachment.

We appreciate for Reviewers’ warm work earnestly, and hope that the correction will meet with approval. Should you have any questions, please contact us without hesitate. 

Once again, thank you very much for your comments and suggestions.

Yours Sincerely,

Zhao Yulin

Round 2

Reviewer 1 Report

Comments and Suggestions for Authors

Figure 1 is very unclear.

Avoid abbreviations like CCECSF .

Abstract,  can provide scientific guidance, this is a bit strange.

Keyword: Public- Private should be Public-Private

Add citation for "In addition, due to the insufficient status of China's pension system, relying solely on the government is inadequate to sustain the operation of CCECSF. " paragraph.

post- evaluation should be post-evaluation

Revise word spacing of "linear regression model[26], support vector machine
[27],neural network class model[28] and machine learning[29]."

Too many abbreviations in the paragraph "The common multi-attribute decision-making models include ELECTRE method[30], PROMETHEE method[31]"

Equation 19 wrong font size.

In order to effectively evaluate political performance, three experts are selected. Who are these experts?

What is sensitivity analysis? Please state: A bi-objective optimization model for the medical supplies' simultaneous pickup and delivery with drones Y Shi, Computers & Industrial Engineering, 2022 - Elsevier

A comparative life-cycle assessment of hydro-, nuclear and wind power: A China study L Wang, Applied Energy, 2019

Government Management Expert etc should state their years of experience etc.

Expert usually use method like AHP: Ranking of risks for existing and new building works, FF Zeng - Sustainability, 2019, what is your method's merit as compared to method like AHP and others?

Section 4 needs citations.

Section 3 needs citations.

Comments on the Quality of English Language

Polish English

Author Response

Dear reviewer:

We are very grateful to your comments for the manuscript. According with your advice, we tried our best to amend the relevant part and made some changes in the manuscript. All your questions have been answered in detail in the attachment.

We appreciate for Reviewers’ warm work earnestly, and hope that the correction will meet with approval. Should you have any questions, please contact us without hesitate. 

Once again, thank you very much for your comments and suggestions.

Yours Sincerely,

Zhao Yulin

尊敬的审稿人:

我们非常感谢您对稿件的评论。根据您的建议,我们尽力修改了相关部分,并对稿件进行了一些修改。您的所有问题均已在附件中得到详细解答。

我们衷心感谢审稿人的热情工作,并希望更正能得到批准。如果您有任何问题,请随时与我们联系。

再次非常感谢您的意见和建议。

此致

赵玉林

Reviewer 2 Report

Comments and Suggestions for Authors

The authors have provided the comments and I have no further comments.

Author Response

尊敬的审稿人:

我们非常感谢您对稿件的评论。我们真诚地感谢审稿人的热情工作。如果您有任何问题,请随时与我们联系。再次非常感谢您的意见和建议。

此致

赵玉林

Round 3

Reviewer 1 Report

Comments and Suggestions for Authors

[J] Should be removed from the references.

Comments on the Quality of English Language

Polish English